# The Role of Cellular Prion Protein in Glioma Tumorigenesis Could Be through the Autophagic Mechanisms: A Narrative Review

**DOI:** 10.3390/ijms24021405

**Published:** 2023-01-11

**Authors:** Daniele Armocida, Carla Letizia Busceti, Francesca Biagioni, Francesco Fornai, Alessandro Frati

**Affiliations:** 1Department of Human Neuroscience, Sapienza University of Rome, Via Caserta 6, 00161 Roma, Italy; 2Department of Oral and Maxillofacial Sciences, Sapienza University of Rome, Via Caserta 6, 00161 Roma, Italy; 3Istituto di Ricovero e Cura a Carattere Scientifico (I.R.C.C.S.) Neuromed, Via Atinense 18, 86077 Pozzilli, Italy

**Keywords:** prion protein, PRNP, cellular prion protein (PrPC), autophagia

## Abstract

The carcinogenesis of glial tumors appears complex because of the many genetic and epigenetic phenomena involved. Among these, cellular prion protein (PrPC) is considered a key factor in cell-death resistance and important aspect implicated in tumorigenesis. Autophagy also plays an important role in cell death in various pathological conditions. These two cellular phenomena are related and share the same activation by specific alterations in the cellular microenvironment. Furthermore, there is an interdependence between autophagy and prion activity in glioma tumorigenesis. Glioma is one of the most aggressive known cancers, and the fact that such poorly studied processes as autophagy and PrPC activity are so strongly involved in its carcinogenesis suggests that by better understanding their interaction, more can be understood about its origin and treatment. Few studies in the literature relate these two cellular phenomena, much less try to explain their combined activity and role in glioma carcinogenesis. In this study, we explored the recent findings on the molecular mechanism and regulation pathways of autophagy, examining the role of PrPC in autophagy processes and how they may play a central role in glioma tumorigenesis. Among the many molecular interactions that PrP physiologically performs, it appears that processes shared with autophagy activity are those most implicated in glial tumor carcinogeneses such as activity on MAP kinases, PI3K, and mTOR. This work can be supportive and valuable as a basis for further future studies on this topic.

## 1. Introduction

Cellular prion protein (PrPC) in its misfolded form is considered the key to the pathogenesis of prion diseases, but in its physiological form is also considered a key factor in cell-death resistance and important aspects implicated in tumorigenesis [1,2,3].

Autophagy also plays an important role in cell death in various pathological and physiological conditions [4]. Autophagy protects cells against death under conditions of starvation and neurodegeneration [5]. These two cellular phenomena are recently considered in a relationship and share the same activation by specific alterations in the cellular microenvironment [6,7,8]. Furthermore, there is an interdependence between autophagy and prion activity in glioma tumorigenesis [9,10]. Because cells in which the prion protein gene (PRNP) was eliminated are more susceptible to apoptotic cell death by serum deprivation and because autophagy is also induced by serum deprivation in various cell lines, it was demonstrated that PrPC is involved in the autophagy pathway [7]. Glioma is one of the most aggressive known cancers [11] of which there is currently no cure [12]. The fact that such poorly studied processes as autophagy and PrPC activity are so strongly involved in its carcinogenesis [13] suggests that by better understanding their interaction, more can be understood about its origin and treatment. Yet few studies exist that relate these two cellular phenomena, much less try to explain their combined activity and role in glioma carcinogenesis. In this study, we address by narrative review the respective roles of PrPC, autophagy, and their combined activities in glioma biology [10]. To clarify the correlation between autophagy and physiologic prion activity, we illustrated the recent findings on the molecular mechanism and regulation pathways of autophagy, exploring and examining the role of PrPC in autophagy processes and how they may play a central role in glioma tumorigenesis. We believe this work can be supportive and valuable as a basis for future studies on this topic.

## 2. The Role of Prion Protein

PrPC is a cell surface glycoprotein, highly conserved in all mammalian species, encoded by the PRNP gene [11]. PrPC is considered the key in the pathogenesis of prion diseases, in which the fundamental event is it’s misfolding into a protease-insensitive, amyloidogenic isoform (PrPSc). Physiologically PrPC acts as a scaffold protein, assembling signaling platforms on the plasma membrane to elicit several biological processes in various cells type, including stem cells [12].

The prion gene family comprises four genes, PRNP, PRND, SPRN, and PRNT, located within a 55 kb region on the 20p13 locus in humans [13]. Prion diseases are caused by the aberrant processing of the everyday cellular PrPC into a disease-causing isoform, denoted PrPSc, where its accumulation results in neurologic dysfunction [14]. In humans, PrPC is expressed in various peripheral tissues and, to a higher extent, in the central nervous system (CNS) [15]. PrPC is physiologically expressed within the CNS, although its content varies among distinct brain regions, cell types, and neurons with different neurochemical phenotypes [9]. Various functions have been proposed for PrPC, including involvement in cell survival, oxidative stress, immunomodulation, differentiation, metal ion trafficking, cell adhesion, and transmembrane signaling. One of the most significant advances has been recognizing that PrPC leads to resistance to cell death, particularly apoptotic cell death, an essential aspect of tumorigenesis, and the development of resistance to drugs used to treat cancer [14]. The prion cancer research field has progressively expanded in the last few years (Table 1). It has yielded consistent evidence for the involvement of PrPC in cancer cell proliferation, migration, invasion, and cancer stem cell properties [3]. Most recent data have uncovered new facets of the biology of PrPC in cancer, ranging from its control of enzymes involved in immune tolerance to its radio-protective activity by promoting angiogenesis [15]. Although the physiological role of PrPC remains fully established, it is considered a key factor for resistance to cell death, particularly to apoptotic cell death, an essential aspect of both tumorigenesis and the development of resistance to drugs used to treat cancer [14,15].

Still, a single dysfunction or alteration of PrPC cannot justify and support the complex process of carcinogenesis that occurs in brain gliomas. PRNP knockout (KO) experimental models have provided some insights into the physiological function of “prion-like” but did not evidence direct alterations. This factor indicates that other molecules can physiologically compensate the PrPC loss of function [26]. Despite this finding, by analyzing only the activity in cell physiology, it seems that PrPC protects neurons against cell death and oxidative stress, controls copper metabolism, synaptic transmission, and cell adhesion, and seems to be involved in maintaining the cellular cycle to activate the immune response [5]. Interestingly, more recent studies suggested that PrPC plays a role in pluripotency and differentiation of embryonic stem cells, cell proliferation, and differentiation through the direct activation of the Src-family kinase Fyn, at least as far as the CNS effects. Starting from these observations, PrPC has been intriguingly involved in the development of human tumors, including glioblastoma (GB) [20,27], and gastric, breast, prostate, and colorectal carcinomas [28].

The role of PrPC in cell differentiation has initiated a dedicated line of research concerning the relationship with totipotent brain cells, the so-called “Glioblastoma stem cells” (GSCs) from which the primary genetic and subsequently epigenetic errors that result in tumor formation are hypothesized to originate [29]. The concentration of PrPC in GSCs depends on an equilibrium between its biosynthesis and degradation rates, which in turn are controlled by a multitude of processes–from mRNA transcription to co-translational secretion into the endoplasmic reticulum, quality control of folding, glycolipid bonding, and vescicular transport, as well as extracellular shedding [25]. Our literature research shows that the most critical works on this topic mainly concern GB cell cultures (generally used since they exhibit a high growth rate in vitro) in which some molecular cascades are identified concerning heat shock protein 70 (Hsp70) Hsp70, heat shock protein 90 (Hsp90), STI1 and various mechanisms involved in the cellular stress and hypoxia phases. Furthermore, the correlation between PrPC and GSC was demonstrated by demonstrating that the levels of their marker proteins [30], such as Oct4, Nanog, Sox2, and ALDH1A1, significantly increased in cancer tissues [9,31]. It is from these recent findings that a central role of cellular autophagy mechanisms has begun to be hypothesized. The fact that intrigued us most during the literature search was that many of these processes in which PrPC is involved turn out to be in close sharing with the known mechanisms of autophagy.

## 3. The Role of Autophagy

Autophagy, also called programmed cell death II, is a term that defines any intracellular process that results in the degradation of cytosolic ingredients inside lysosomes [31]. Autophagy is a major pathway of degradation of cytoplasmic constituents and cell death by necrosis and apoptosis. It was initially described as a cellular response to nutrient starvation, thus facilitating cell survival via the degradation and recycling of cytoplasmic contents. The typical morphological feature is the formation of many large autophagic vacuoles (AVs). Although its specific role is still controversial, the participation of the lysosomal system in programmed cell death has recently been well accepted [32]. Autophagy plays an important role in cell survival and cell death in various pathological and physiological conditions. The lysosomal system involved in autophagy is especially relevant in remodeling the developing organs and several pathological conditions such as hypoxia, ischemia, inflammation, and exposure to toxic agents [15,33]. By eliminating intracellular damaged organelles and aggregates, autophagy promotes cell surface antigen presentation and cellular senescence, protects against genome instability, and prevents necrosis, offering it an essential role in preventing diseases (Figure 1) such as neurodegeneration, cardiomyopathy, diabetes, liver disease, autoimmune diseases, infections, and cancer [33]. In the CNS, autophagy protects cells against death under conditions of starvation and neurodegeneration but is also an important contributing factor in some types of cell death [5]. The increased number of AVs in neurons is a frequent observation in many neurodegenerative diseases. Indeed, the most recent research involves diseases such as Alzheimer’s syndrome, Parkinson’s syndrome, and the more common forms of frontotemporal dementia. It seems that neuronal autophagy plays a critical role at basal physiological levels in controlling intracellular quality and maintaining nervous system health by removing aggregated proteins. At first, it was believed that neuronal autophagy is relatively inactive, but in recent years genetic research using mouse models emphasized the importance of autophagy in the non-proliferating cells in neurons [34].

It is important to highlight that in cancer cells, acute amino acid starvation induces three distinct pathways of active autophagy: mTOR inactivation with subsequent activation of chaperone-mediated autophagy (CMA), induction of an immediate selective endosomal microautophagy independent of mTORC1 inactivation and canonical macroautophagy. These autophagic pathways likely work in concert and may partially compensate for each other.

Recently, the concept of the consequences of increased CMA in the CNS, driven by GB cells, as a critical player iFn GB progression has been explored. CMA is a selective autophagy pathway that degrades specific proteins in lysosomes. Proteins are directly recognized one by one by the chaperone Hsp70, although other chaperones can be found in the Hsp70-substrate protein complex. During CMA, the substrate protein binds to Hsp70, and lysosomal-associated membrane protein 2A (LAMP-2A) at the lysosomal membrane to be unfolded, translocated, and degraded within the lysosome with the participation of another chaperone Hsp70, which resides within the lysosome lumen [30,31,32]. CMA activity is highly regulated by cell context and, thereby, the stimuli the cell receives, as these modulate the amount of functional LAMP-2A that is mobilized to the lysosomal membrane [35]. While physiological CMA levels are needed for cell homeostasis and to avoid malignant cell transformation by preventing DNA damage and increasing proteostasis, CMA activity becomes pro-oncogenic when dysregulated [36]. Transformed cells abnormally upregulate CMA activity, which degrades different tumor cell survival- and proliferation-inhibitory proteins. In addition, increased CMA is required to sustain the characteristic Warburg metabolism in tumor cells as it is essential to degrade the majority of glycolytic enzymes and, therefore, to induce anaerobic glycolysis needed for cancer progression [18].

Macroautophagy is a process of lysosomal degradation and recycling of cellular components by sequestering cytosolic regions in de-novo-generated double-membrane vesicles knowns as autophagosomes. This process can be either in bulk or selective. The selectivity is mediated by autophagy receptors that bring cargo to the phagophore, including damaged organelles, intracellular bacterial pathogens, and aggregated proteins.

Under normal conditions, the basal level of autophagy is very low. Therefore, autophagy must be induced through an efficient mechanism to survive stress and adapt to extracellular signals. In mammalian cells under nutrient-rich conditions, autophagy is inhibited by the serine/threonine-protein kinase mTOR (mammalian target of rapamycin). As previously described by our research group [37], GB can be considered at least in the early stages of its carcinogenesis, a biologically dependent pathology with a defect in mTOR activity. mTOR negatively regulates other serine/threonine kinases, Unc-51-like kinase-1 (ULK1) and -2 (ULK2), phosphorylating and inactivating Unc-51-like kinases (ULKs) and autophagy-related gene 13 (Atg13) by binding to the ULKs-Atg13-FIP200 complex. Under nutrient stress conditions, AMP-dependent protein kinase (AMPK) increases autophagy by inhibiting mTOR complex 1 (mTORC1) via phosphorylation. Then, ULK1 and ULK2 are activated and phosphorylate Atg13 and FIP200. The beclin1 complex is recruited and activates the class III PI3K VPS34, stimulating autophagosome nucleation.

Autophagy is also induced by Exchange protein directly activated by cAMP 1 (Epac-1) through a Ca^2+^/calmodulin-dependent kinase b (CaMKKb)/AMPK signaling pathway. Recently, inositol polyphosphate multikinase (IPMK) has been reported to induce autophagy by regulating AMPK/ULK activation and enhancing autophagy-related transcription-dependent of Sirt-1 activation. In tumor cells, many signaling pathways participate in the regulation of autophagy. Down-regulation of STAT/BCL2/BECLIN-1 and PI3K/Akt/mTOR-mediated signaling pathways can activate macroautophagy. The Ras/Raf/ERK signaling pathway is one of the most frequent pathways that activate tumor macroautophagy. Other important modulators of macroautophagy in GB are Sirt1, that induces autophagic cell death and mitophagy, insulin, p53, p38 mitogen-activated protein kinase (p38-MAPK), five ′ AMPK, phosphatase and tensin homolog deleted from chromosome 10 (PTEN), and reactive oxygen species (ROS)-associated pathways. Furthermore, intracellular calcium signaling seems also an essential regulatory pathway in GB. Recent advances in Hedgehog, NRF2-P62, and PD-L1/PD1 signaling pathways appear to indicate that these might be potential targets to modulate macroautophagy in glioma.

## 4. The Shared Activity in Glioma Tumorigenesis

PrPC has been intriguingly involved in developing human tumors, including gliomas gastric, breast, prostate, and colorectal carcinomas [18,24,35,36,37]. The discovery of PrPC expression in different types of stem cells and evidence on PrPC overexpression in a variety of tumors has recently prompted its investigation in GSC research [10,38]. PrPC is highly expressed within human GB cell lines [10,16,17,22,38].

To fully understand the role PrPC plays in glioma tumorigenesis, it is necessary to identify its functions within cellular activities. Still, at present, this represents one of the main questions to be resolved. Indeed, PRNP-knockout experiments did not evidence particular alterations in mice, indicating that PrPC is not essential for normal development or that PrPC loss of function can be compensated by other molecules [22]. The question that arises is, at this point, how is it possible that a protein that physiologically turns out to be superfluous in cellular functioning can be repeatedly implicated in the hypotheses of tumor carcinogenesis in such varied and diverse forms [19]. Tumor carcinogenesis involves more than 570 protein interactions with PrPC defining it as what has recently been described as PrpC-connectome [39]. Most of these molecules play a crucial role in signal-transducing events essential for neuronal function. However, none could serve as the chaperone ‘protein X’ [22]. One of the processes that can be considered a common denominator in all PrPC interactions may be its role and activity in autophagy processes [22]. Autophagy is an evolutionarily conserved catabolic pathway involved in the turnover of proteins, protein complexes, and organelles through lysosomal degradation. Because neurons are postmitotic and do not replicate in general, they are heavily dependent on basal autophagy compared with non-neuronal cells, as misfolded proteins inside neurons cannot be diluted through cell division [40].

The activity of PrPC and autophagy mechanisms converge in several processes responsible for glioma tumorigenesis [41,42,43,44]. It was recently discovered that some human prion peptides induce autophagy flux and autophagic cell death in the neuronal cell [23,45,46]. Further, autophagy inhibitors can reduce prion protein-induced autophagic cell death via AMP pathway [47,48,49]. An increased calcineurin activity mediated by prion protein-induced neuronal cell damage and provided insight into the fundamental mechanism underlying AMPK and autophagy flux in prion disease [47].

AMPK is a sensor of energy status, including energy deficiency based on the increased AMP/ATP ratio. Calcineurin prolonged the oxidative stress-induced activation of calcineurin resulting in the attenuation of AMPK signaling and inhibition of autophagy. It was shown that prion protein-mediated calcineurin activation decreased the AMPK activity via dephosphorylation [48] and suggests that AMPK inactivation may lead to autophagy activation and neuronal toxicity.

So, it was demonstrated that prion protein-calcineurin activation was involved in prion protein-mediated neuronal cell death and AMPK and autophagy signaling pathways [49]. It regulates metabolic homeostasis by controlling autophagy [8]. It seems that prion peptide decreased the AMPK activity via dephosphorylation and increases autophagic cell death [49]. At the same time, prion protein increases calcineurin activity, resulting in decreased AMPK phosphorylation that induces autophagic cell death [47].

Other interesting factors discovered in this finding were that PrP silencing in glioma cell lines causes increased autophagy due to induction of LC3-II, an increase in Beclin 1, and simultaneous decreases in p62, Bcl-2, and the phosphorylation of 4E-BP1, a target of mTOR autophagy signaling [47]. Interestingly, mTOR, a master player in cell signaling with a pivotal role in tumorigenesis, is also involved in PrPC-dependent neuronal differentiation and neuroprotection through the activation of PI3K/Akt pathways [21]. Indeed, one of the main keys in these pathways through a shared process in which both autophagy and PrPC-related processes participate is Hsp-70. During CMA, the substrate protein binds to Hsp70 at the lysosomal membrane to be unfolded, translocated, and degraded within the lysosome with the participation of another chaperone, Hsp70, which resides within the lysosome lumen [30,31,32].

In addition, it was reported that the pharmacological induction of autophagy by treatment with trehalose or lithium could decrease pathogenic and infectious PrPSc expression in persistently prion-infected neurons. Autophagy is involved in cell survival in response to nutrient deprivation and is associated with various diseases [41,42]. There are many processes in common between neurodegenerative diseases and the more aggressive forms of glial tumors both from a biomolecular point of view and then from a pathological and clinical point of view, so considering known pathogenic processes in some prion diseases can, in our opinion help in understanding an equally complex pathology such as glioma [43].

In earlier findings, it has been reported that the number of autophagosomes and other pre-lysosomal vesicles significantly increase in neurodegenerative disorders [4]. Some evidence showed that autophagic dysfunction is linked to aging and diseases, including cancer and neurodegenerative disorders.

From what is found so far, there is good experimental evidence from in vivo and in vitro studies demonstrating that autophagy, as a housekeeper in prion diseases, when induced by chemical compounds, can play a protective role in the clearance of pathological PrPSc accumulated within neurons. When medicines promote autophagy by simultaneously activating both mTOR- dependent and -independent pathways, it showed more significance than the maximum effect of one pathway alone. In contrast, defective autophagy may lead to the occurrence of neurodegenerative diseases and contribute to the formation of spongiform changes.

The neuroprotective effect of autophagy processes is known and shared in many degenerative-type diseases, each of which shares some mechanisms with processes governed by PrPC activity, examples being the role of α7nAChR in Alzheimer’s disease [41], LC3 in Parkinson’s dementia [40], and colorectal cancer. Further, the presence of AVs has been found within pre-synaptic terminals in brains of patients with prion disease [44]. In addition, ultrastructural localization of scrapie-responsive gene 1 (Scrg1) protein was associated with AVs in the central neurons of scrapie-infected mice.

AVs develop not only in the neuronal perikarya but also in neuronal processes, eventually replacing the cross-section of affected neurites. In Neuroblastoma cells transfected in vitro with three different prion protein mutants (V203I, E211Q, and Q212P), all three protein mutants were converted into PrPSc -like form and accumulated in aggresomes [46]. As a conserved host defense response to infection, autophagy plays a protective role in prion diseases by degrading aggregate-prone proteins accumulated within endosomal/lysosomal vesicles. Recent studies provided the first direct evidence that autophagy induction results in cellular PrPSc degradation. When autophagy is suppressed by pharmacological interference or siRNA gene-silencing of essential members of the autophagic machinery, the capacity of compound-induced autophagy in reducing cellular levels of PrPC is impaired [49,50,51] (Figure 1). Because cells in which the PRNP was eliminated are more susceptible to apoptotic cell death by serum deprivation. Because autophagy is also induced by serum deprivation in various cell lines, it was explored whether PrPC is involved in the autophagy pathway [45].

Additionally, the knockdown of PrPC expression in various cancer cells increases autophagy-mediated cell death. However, one study suggested that PrPC has a neuroprotective effect associated with the induction of autophagy against oxidative stress in hippocampal neuron cells. Moreover, Shin et al. [41] found that the increase in autophagic flux caused by the depletion of PrPC is correlated with age in the hippocampus compared to that in normal mice [42,45]. The same study suggested that a PrPC deficiency may disrupt autophagic flux by blocking autophagosome-lysosomal fusion. These results suggest that PrPC is a key factor in regulating autophagic flux in the brain, although the relationship between PrPC and autophagic flux is unclear [39,46].

## 5. Further Studies Needed

In the cellular processes that are shared by autophagy mechanisms and PrPC many in the studies here are only mentioned and not explored in depth with ad-hoc studies. For example, in the Hsp-70 reports there seems to be an important link with STI1 activity, which is recognized as essential in the carcinogenic process of many forms of cancer. In fact, it is recognized that STI1is secreted by a GB cell line and induces proliferation of distinct glioma cell lines; further, the Erk and Akt signaling pathways mediate STI1-induced proliferation.

In addition, treatment with STI1 activated Erk and Akt, indicating the involvement of these signaling pathways in the proliferative effect. Interestingly, treatment with LY294002 induced an increase in the phosphorylation levels of Erk, a finding that suggests the existence of crosstalk between these pathways. STI1 imposes a small and transient Erk activation, leading to increased proliferation. On the other hand, when PI3K was inhibited, STI1- induced activation of Erk was more intense and durable, possibly because, in this situation, Akt did not counterbalance erk pathway activity. This pattern of Erk activation may cause cell cycle arrest. Cells subjected to treatment solely with LY294002 or STI1 showed a similar increase in Erk activation as assayed after a 5-min treatment. However, as opposed to STI1-treated cells, the PI3K inhibitor induced a much more durable Erk activation, which persisted for at least 1 h and was unrelated to an increase in proliferation. Together, these data indicate that parallel activation of both the Erk and Akt pathways are required for the proliferative effects of STI1 and that the intensity and duration of Erk activation may ultimately determine the final impact of STI1 upon proliferation. Further studies are needed to this aspect.

Direct therapy against this glioma is very complicated due to GB’s intrinsic cellular heterogeneity and the potential adverse effects that this intervention might have on peritumoral cells [52,53]. Pharmacological manipulation of CMA might be used as a specific therapy against GB growth. Inhibition of CMA would directly impact GB cells and other tumor-related cell types, such as PCs, which help the tumor spread in the brain. However, specific inhibitors against LAMP-2A or CMA modulators have not been found or demonstrated to work. Gene therapy directed to modulate CMA in PCs specifically should be cautiously considered as an alternative, as physiological CMA in PCs is essential to maintain healthy brain homeostasis.

Tumor cells wickedly manipulate and increase CMA in PCs to benefit from changes in the tumor microenvironment (TME). This can be seen as an opportunity to develop new pharmacological therapies countering this process to treat GB. The development of such treatments might also be helpful in the treatment of other highly vascularized cancer types where peritumoral pericytes (PCs) show a pro-tumoral regulatory role. Generalized CMA blockage in peritumoral brain cells (including PCs and probably other immune cells) should lead to a pro-inflammatory niche capable of preventing tumor progression and promoting anti-tumor immune responses to eliminate GB cells [23]. However, further studies are needed to understand its actual clinical role in the progression of brain gliomas.

## 6. Conclusions

There are common mechanisms between autophagy and PrPC activity actively involved in the development and progression of tumor pathology, especially in gliomas. It was recently discovered that some human prion peptides induce autophagy flux and autophagic cell death in neuronal cells, and these reported mechanisms could be implicated in glioma tumorigenesis. Further, autophagy inhibitors can reduce prion protein-induced autophagic cell death via AMP pathway. An increased calcineurin activity mediated by prion protein-induced neuronal cell damage provided insight into the fundamental mechanism underlying the AMPK and autophagy flux in gliomas.

## Figures and Tables

**Figure 1 ijms-24-01405-f001:**
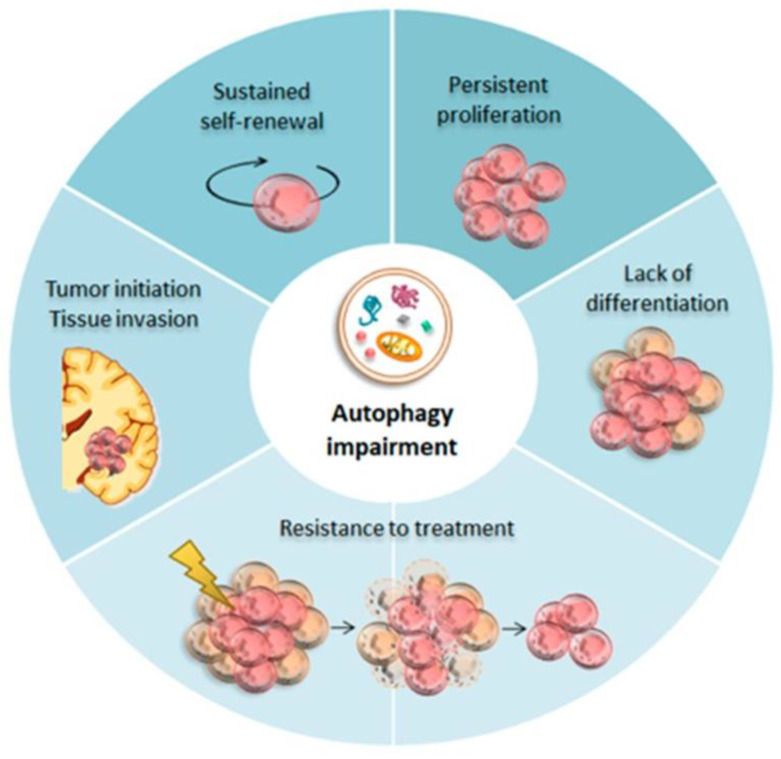
Autophagy in Glioma Stem Cells (GSCs). Baseline autophagy guarantees neuronal differentiation and homeostasis within the neural stem cells (NSCs) niche during development. Impaired autophagy seems to be crucial for GSCs tumor initiation.

**Table 1 ijms-24-01405-t001:** Table reports some of the most important research-line on the role of PrPC in glioma tumorigenesis.

No	Authors		Analysis	Object
1	Kikuchi, Y., et al. (2002) [2]	Human glioblastoma cell line T98G	Immunoblot analysis	PrP C rilevation
2	Comincini, S., et al. (2004) [16]	Human glioblastoma cell line T98G	Western blot and immunohistochemistry	PrP C rilevation
3	Kikuchi, Y., et al. (2004) [1]	Human glioblastoma cell line T98G	Immunoblot analysis	PrP res production
4	Comincini, S., et al. (2007) [17]	Human tissue samples	m-RNA analysis and PCR	Prnd doppel rilevation
5	Erlich, R. B., et al. (2007) [18]	human glioblastoma-derived cell line A172	thymidine incorporation assays	co-chaperone stress-inducible protein 1 (STI1) as a cell surface ligand for cellular prion (PrP(C) producted
6	Kikuchi, Y., et al. (2008) [7]	Human glioblastoma cell line T98G	m-RNA analysis and PCR	GPI-anchorless PrP (GPI(-) PrPSV) producted in hypoxia
7	Sbalchiero, E., et al. (2008) [19]	two astrocytoma-derived human cell lines (IPDDC-A2 and D384-MG)	double-immunofluorescence staining and confocal microscopy	doppel encoding gene (PRND) is over-expressed and the corresponding protein product (Dpl) is ectopically localized in the cytoplasm of the tumor cells
8	Fonseca, A. C., et al. (2012) [6]	human glioblastoma cell line GBM95	Immunoblot analysis	STI1 is secreted by microglia and favors tumor growth and invasion through the participation of MMP-9 in a PrP(C)-independent manner
10	Lopes, M. H., et al. (2015) [20]	Fresh surgical astrocytoma and non-neoplastic brain tissue samples	qPCR	heat-shock protein 70 (Hsp70)-Hsp90-organizing protein (HOP). PrP(C)-HOP engagement is a promising approach for GBM therapy.
11	Santos, T. G., et al. (2015) [21]	Review	Review	PrP(C)-organized multicomplexes
12	Corsaro, A., et al. (2016) [22]	four GBM CSC-enriched cultures, from Human tissue specimen	Western blot and immunohistochemistry, ELISA	PrPC controls the stemness properties of human GBM CSCs and that its down-r
13	Iglesia, R. P., et al. (2017) [23]	glioblastoma stem-like cells cultured in neurospheres with growth factors.	immunofluorescence and flow cytometry	co-chaperone Hsp70/90 organizing protein (HOP)
14	Yang, X., et al. (2017) [24]	Book chapter	Using PDAC cell lines BxPC-3 and AsPC-1 as model system	dysfunction of glycosylphosphatidylinositol (GPI) anchor synthesis machinery resulted in the biogenesis of pro-PrP exacerbating tumorigenesis.
15	Kim, Y. C., et al. (2020) [3]	Cancer Genome Atlas (TCGA) database	silico analysis using PolyPhen-2, PANTHER, PROVEAN, and AMYCO	48 somatic mutations in the PRNP gene, including 8 somatic mutations that are known pathogenic mutations of prion diseases
16	Heinzer, D., et al. (2021) [25]	human U251-MG glioblastoma cells	miRNA, siRNA	The RNA-binding post-transcriptional repressor Pumilio-1 was identified as a potent limiter of PrPC expression through the degradation of PRNP mRNA

## Data Availability

Not applicable.

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
