# Peer review of "The Role of Cellular Prion Protein in Glioma Tumorigenesis Could Be through the Autophagic Mechanisms: A Narrative Review"

_ijms, 2023, doi:10.3390/ijms24021405_

Round 1
Reviewer 1 Report
Authors illustrated the recent findings on the molecular mechanism and regulation pathways of autophagy, investigating the role of PrPc in this processes and how it is related to the glioma tumorigenesis. Authors have focused attention on how common mechanisms between autophagy and PrPc activity may be actively involved in the development and progression of tumor pathology, especially in gliomas. This work can be useful for further future studies on this topic.
The manuscript should be checked for errors. In the abstract (24th line), it seems there is a missing word: "In this study we ... the recent".
Author Response
Response to reviewer 1
Authors illustrated the recent findings on the molecular mechanism and regulation pathways of autophagy, investigating the role of PrPc in this processes and how it is related to the glioma tumorigenesis. Authors have focused attention on how common mechanisms between autophagy and PrPc activity may be actively involved in the development and progression of tumor pathology, especially in gliomas. This work can be useful for further future studies on this topic.
The manuscript should be checked for errors. In the abstract (24th line), it seems there is a missing word: "In this study we ... the recent”.
R: We would like to thank the reviewer for his appreciation of our work and for fully framing the objective of this study. We have made the requested changes, revising and editing the abstract.
We performed a new grammar check using a professional English editing service and revised all abbreviations to make it easier to understand.

Reviewer 2 Report
The manuscript intended to review current knowledge on cellular prion protein role in glioblastoma tumorigenesis with special reference to autophagia. However, prior to publishing some major issues need to be addressed, concerning the structure and composition of the article, to avoid repetition and ambiguity. Furthermore, to ensure consistency, all abbreviations need to be explained when first mentioned in the text, and used in the same form throughout. Conclusions should be composed on what is stated previously, in line with major issues discussed. Special attention should be given to the English grammar and changes made accordingly.
Specific comments:
Examples of grammar errors
Line 24 : In this study we the recent findings on the molecular mechanism and regulation pathways of autophagy
Line 25 exploring and examines the role of PrPc in autophagy processes and how they may play a central role in glioma tumor- 26 igenesis.
Line 29 This work can be supportive and useful as a basis for further future studies on this topic.
Lines 64-65: to elicit several biological processes, including stem cells [ …which processes?)
Examples of abbreviation and other inconsistencies:
Lines 45, 47, 90 - PrPC, or PrP?
Line 43, 60: Prnp or PRNP?
Lines 35 and 68: of the everyday cellular or in its physiological
Line 115 CSC or GSC
Line 155- GB-driven increase (explain abbreviation)
Line247: Prp-connectoma (explain abbreviation)
Line 269, 275: prion protein is the same as PrPc or PrP?
Line 310: the presence of Avs (abbreviation meaning)
line 373-374: Tumor cells wickedly manipulate and increase CMA (abbreviation meaning?) in PCs to benefit from changes in the TME,
Lines 339-343: In the Conclusion ST1 is mentioned for the first time. Rephrase and explain, transfer to other section in the text.
“This investigation showed that (1) STI1 is secreted by a glioblastoma cell line; (2) STI1 induces proliferation of distinct glioma cell lines; (3) the Erk and Akt signaling pathways mediate STI1-induced proliferation; (4) STI1 does not induce proliferation in normal as- trocytes; and (5) STI1-induced proliferation of A172 cells depends on its PrPC binding domain.”
Author Response
Response to Reviewer 2
The manuscript intended to review current knowledge on cellular prion protein role in glioblastoma tumorigenesis with special reference to autophagia.
R: We first want to thank the reviewer for carefully reviewing our manuscript and grasping the key message we wanted to bring to the publication. Below we submit the point-to-point responses.
However, prior to publishing some major issues need to be addressed, concerning the structure and composition of the article, to avoid repetition and ambiguity.
Furthermore, to ensure consistency, all abbreviations need to be explained when first mentioned in the text, and used in the same form throughout.
R: We have re-organized the key information in the various sub-heads to make the text more linear and usable. Abbreviations have been brought back in order of appearance and have been standardized throughout the text.
Conclusions should be composed on what is stated previously, in line with major issues discussed.
R: The conclusions have been rewritten and appear more concise and speculative toward the information in the discussion section
Special attention should be given to the English grammar and changes made accordingly.
Specific comments:
Examples of grammar errors
Line 24 : In this study we the recent findings on the molecular mechanism and regulation pathways of autophagy
R: corrected
Line 25 exploring and examines the role of PrPc in autophagy processes and how they may play a central role in glioma tumor- 26 igenesis.
R: corrected
Line 29 This work can be supportive and useful as a basis for further future studies on this topic.
R: corrected
Lines 64-65: to elicit several biological processes, including stem cells [ …which processes?)
R: corrected
Examples of abbreviation and other inconsistencies:
Lines 45, 47, 90 - PrPC, or PrP?
Line 43, 60: Prnp or PRNP?
R: we have standardized the abbreviations by defining PrPC as the cellular form of the prion protein and PRP as the gene encoding it.
Lines 35 and 68: of the everyday cellular or in its physiological
R: corrected
Line 115 CSC or GSC
R: Is GSC in any abbreviation: corrected
Line 155- GB-driven increase (explain abbreviation)
R: corrected
Line247: Prp-connectoma (explain abbreviation)
R: corrected
Line 269, 275: prion protein is the same as PrPc or PrP?
R: corrected and explained above
Line 310: the presence of Avs (abbreviation meaning)
R: AVs means autophagic vacuoles (explained the abbreviation in the text)
line 373-374: Tumor cells wickedly manipulate and increase CMA (abbreviation meaning?) in PCs to benefit from changes in the TME,
R: corrected
Lines 339-343: In the Conclusion ST1 is mentioned for the first time. Rephrase and explain, transfer to other section in the text.
“This investigation showed that (1) STI1 is secreted by a glioblastoma cell line; (2) STI1 induces proliferation of distinct glioma cell lines; (3) the Erk and Akt signaling pathways mediate STI1-induced proliferation; (4) STI1 does not induce proliferation in normal as- trocytes; and (5) STI1-induced proliferation of A172 cells depends on its PrPC binding domain.”
R: we re-writed the conclusion section

Round 2
Reviewer 2 Report
Thank you for the corrections and improvements of the text.
I have no further requests.